# On the Trajectory of Stochastic Gradient Descent Learning in the Information Plane

## Abstract

Studying the evolution of information theoretic quantities during Stochastic Gradient Descent (SGD) learning of Artificial Neural Networks (ANNs) has gained popularity in recent years. Nevertheless, these type of experiments require estimating mutual information and entropy which becomes intractable for moderately large problems. In this work we propose a framework for understanding SGD learning in the information plane which consists of observing entropy and conditional entropy of the output labels of ANNs. Through experimental results and theoretical justifications it is shown that, under some assumptions, the SGD learning trajectories appear to be similar for different ANN architectures. First, the SGD learning is modeled as a Hidden Markov Process (HMP) whose entropy tends to increase to the maximum. Then, it is shown that the SGD learning trajectory appears to move close to the shortest path between the initial and final joint distributions in the space of probability measures equipped with the total variation metric. Furthermore, it is shown that the trajectory of learning in the information plane can provide an alternative for observing the learning process, with potentially richer information about the learning than the trajectories in training and test error.

## 1 Introduction

How do information theoretic quantities behave during the training of ANNs? This question was addressed by Shwartz-Ziv & Tishby (2017) in an attempt to explain the learning through the lens of the information bottleneck method (Tishby et al., 1999). In that work, the layers of an ANNs are considered random variables forming a Markov chain. The authors constructed a 2D information plane by estimating the mutual information values between hidden layers, inputs, and outputs of ANNs. Using this approach it was observed that the information bottleneck method provides an approximate explanation for SGD learning. In addition, their experiments showed the role of compression in learning. That initial paper motivated further work on this line of research (Saxe et al., 2018; Gabrié et al., 2018). The main practical limitation of that type of experiments is that it requires estimating mutual information between high dimensional continuous random variables. This becomes prohibitive as soon we move to moderately large problems, such as the CIFAR-100 dataset, where the large ANNs are employed. Other works dealing with information theoretic quantities tend to have these experimental limitations. For instance, Russo & Zou (2015); Xu & Raginsky (2017); Asadi et al. (2018) used generic chaining techniques to show that generalization error can be upper bounded by the mutual information between the training dataset and output of the learning algorithm. Nevertheless, estimating that mutual information to verify those results experimentally becomes intractable. Furthermore, in our previous work (Anonymous, 2018) we defined a novel 2D information plane that only requires to estimate information theoretic quantities between the correct and estimated labels. Since these random variables are discrete and one-dimensional, this framework can be used to study learning in large recognition problems as well. Moreover, that work provides a preliminary empirical study on the behavior of those information theoretic quantities during learning along with some connections between error and conditional entropy.

In this work, we extend the experiments from Anonymous (2018) to more general scenarios and aim to characterize the observed behavior of SGD. Our main contributions are as follows:

- We define a 2D-information plane, inspired by the works of Shwartz-Ziv & Tishby (2017), and use it to study the behavior of ANNs during SGD learning. The main quantities are entropy of the output labels and its conditional entropy given true labels.

- It is shown that if the learning is done perfectly and under some other mild assumptions, the entropy tends to increase to its maximum.

- It is additionally shown that SGD learning trajectory follows approximately the shortest path in the space of probability measures equipped with the total variation metric. The shortest path is characterized well by a Markov chain defined on probabilities of estimate labels conditioned on true labels. To that end we provide theoretical and experimental justifications for constructing a simple Markovian model for learning, and compare it with SGD through experiments. These experiments are conducted using various datasets such as MNIST (LeCun et al., 1998), CIFAR-10/ CIFAR-100, spirals (Anonymous, 2018), as well as different ANN architectures like Fully Connected Neural Networks (FCNNs), LeNet-5 (LeCun et al., 1999), and DenseNet (Huang et al., 2017).

- The trajectory, however, is not universal. Through a set of experiments, it is shown that SGD learning trajectory differs significantly for different learning strategies, noisy labels, overfitting, and underfitting. We show examples where this type of trajectories provide a richer view of the learning process than conventional training and test error, which allows us to spot undesired effects such as overfitting and underfitting.

The paper is organized as follows: Section 2 introduces the notation as well as elementary notions from information theory. Section 3 formulates learning as a trajectory on the space of probability measures, defines the notion of shortest learning path, and provides a connection to Markov chains. Section 4 constructs a simple Markov chain model for gradient based learning that moves along the shortest learning path. Finally, Section 5 performs an empirical evaluation of the proposed model.

## 2 SYSTEM MODEL

Let $\mathbf{x} \in \mathbb{X}$ be a random vector belonging to some set $\mathbb{X}$ of possible inputs. We assume that there exists a function, known as "oracle", that maps $\mathbf{x}$ to one of $K \in \mathbb{N}$ classes. Formally, there exists a deterministic mapping $c : \mathbb{X} \to \mathbb{Y}$ where $\mathbb{Y} = \{0, \ldots, K - 1\}$ is the set of possible classes. Then, let $\tilde{\mathbf{y}} \in \mathbb{Y}$ denote the random variable $\tilde{\mathbf{y}} = c(\mathbf{x})$. One common assumption, that is present in popular datasets such as MNIST, CIFAR-10, CIFAR-100, and Imagenet, is that $\tilde{\mathbf{y}}$ is uniformly distributed. We assume this to be true throughout this paper. Note that the designer of the dataset has control over the marginal distribution of $\tilde{\mathbf{y}}$.

We model the effect of having error-prone labels, denoted by the random variable $\mathbf{y}$, in the data by introducing discrete independent random noise $\mathbf{z} \in \mathbb{Y}$ to $\tilde{\mathbf{y}}$ in the form of modulo addition[1], that is $\mathbf{y} = \tilde{\mathbf{y}} \oplus \mathbf{z} \in \mathbb{Y}$. Let $\theta \in \Theta$ be the vector, possibly random, containing of all tunable parameters in the hypothesis space $\Theta$. Then a classifier is a deterministic function $g : \Theta \times \mathbb{X} \to \mathbb{Y}$ that aims to approximate $c$. Further, $\hat{\mathbf{y}} = g(\theta, \mathbf{x})$ is defined to be the random variable of the label predicted by the classifier $g(\theta, \cdot)$. Using this notation we define the three types of error: the dataset error $p = P(\mathbf{y} \neq \tilde{\mathbf{y}})$, the test error $\varepsilon = P(\hat{\mathbf{y}} \neq \mathbf{y})$, the true error $\tilde{\varepsilon} = P(\hat{\mathbf{y}} \neq \tilde{\mathbf{y}})$. A summary of this system model is provided in Figure 1. We shortly review some elementary concepts from information theory such as entropy and mutual information. The entropy of a discrete random variable $\mathbf{y} \in \mathbb{Y}$ is defined as[2]

$$H(\mathbf{y}) = - \sum_{y \in \mathbb{Y}} P(\mathbf{y} = y) \log P(\mathbf{y} = y) = -\mathbb{E} \log P(\mathbf{y}).$$

The entropy is bounded by $0 \leq H(\mathbf{y}) \leq \log |\mathbb{Y}|$ and it measures the amount uncertainty present in $\mathbf{y}$. Similarly, the conditional entropy between two random variables $\mathbf{y}$ and $\hat{\mathbf{y}}$ is

$$H(\mathbf{y}|\hat{\mathbf{y}}) = -\mathbb{E} \log P(\mathbf{y}|\hat{\mathbf{y}})$$

and it quantifies the uncertainty about $\mathbf{y}$ given that $\hat{\mathbf{y}}$ is known. Finally, the mutual information $I(\mathbf{y}; \hat{\mathbf{y}})$ between $\mathbf{y}$ and $\hat{\mathbf{y}}$ measures how much information does one random variable carry about the

---

[1]We use $\oplus$ to denote the modulo $K$ addition.

[2]In this paper we assume $\log$ to be the natural logarithm.

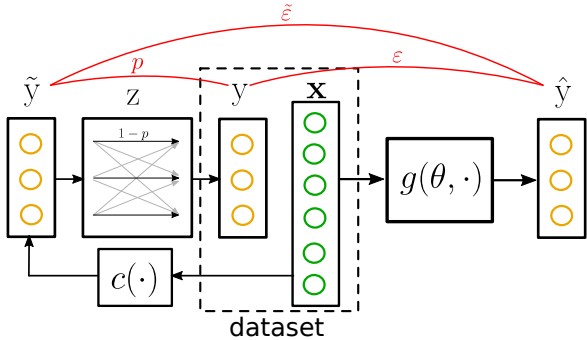

Figure 1: System model.

other. It may be defined in terms of entropies as

$$I(\mathbf{y}; \hat{\mathbf{y}}) = H(\mathbf{y}) - H(\mathbf{y}|\hat{\mathbf{y}}) = H(\hat{\mathbf{y}}) - H(\hat{\mathbf{y}}|\mathbf{y}).$$

Moreover, the following proposition is a well-known result from information theory, known as Fano's inequality, that relates test error $\varepsilon$ and conditional entropy.

**Proposition 1.** *Fano's Inequality (Csiszàr & Körner (2011), Lemma 3.8)*
*The value of $H(\mathbf{y}|\hat{\mathbf{y}})$ and $H(\hat{\mathbf{y}}|\mathbf{y})$ is upper bounded by a function of the expected error as*

$$\max\{H(\mathbf{y}|\hat{\mathbf{y}}), H(\hat{\mathbf{y}}|\mathbf{y})\} \leq \Psi(\varepsilon),$$

*where the function $\Psi : [0, 1] \to \mathbb{R}$ is defined as*

$$\Psi(x) = x \log(K - 1) + h_b(x), \quad x \in [0, 1]$$

*and $h_b(x) = -x \log(x) - (1 - x) \log(1 - x)$ is the binary entropy function.*

This results provides an upper bound on conditional entropy in terms of $\varepsilon$, that is known to be sharp.

In the works of Feder & Merhav (1994) it has been shown that $I(\mathbf{y}; \hat{\mathbf{y}})$ gives an upper and lower bound on the minimal error [3] between y and ŷ. In addition, the minimal error is minimized when $I(\mathbf{y}; \hat{\mathbf{y}})$ reaches its maximum. Therefore, learning can be modeled as finding $\theta$ such that $I(\mathbf{y}; \hat{\mathbf{y}})$ is maximized. This can be written in terms of entropies as

$$\max_{\theta \in \Theta} H(\hat{\mathbf{y}}) - H(\hat{\mathbf{y}}|\mathbf{y}). \tag{1}$$

As in our previous work (Anonymous, 2018) we are interested on characterizing the trajectory in the 2D information plane, composed by $H(\hat{\mathbf{y}})$ and $H(\hat{\mathbf{y}}|\mathbf{y})$, during the learning process of artificial neural networks (ANNs). In Figure 2 we observe that learning trajectory for the DenseNet architecture of 100 layers as it learns to classify data from the CIFAR-100 dataset, for $p = 0$. Intuitively, when solving equation 1, maximizing $H(\hat{\mathbf{y}})$ is more related with the unsupervised component of learning since it does not depend y. On the other hand, keeping $H(\hat{\mathbf{y}}|\mathbf{y})$ low while $H(\hat{\mathbf{y}})$ increases can be seen as the supervised component of equation 1. From this point of view it would be interesting to characterize the inflection point from which $H(\hat{\mathbf{y}}|\mathbf{y})$ starts decreasing, since it allows us to observe at which point SGD starts paying more attention to assigning labels correctly than to learn about the distribution of the input. One also may wonder if this increasing-decreasing trajectory is an accidental result for that occurs only on this particular experimental setup, or if it is a fundamental property of SGD. In Anonymous (2018) we showed that this behavior seems to appear regardless of the activation function employed (see Appendix C) on the spirals and MNIST dataset. Moreover, in further sections we provide a justification for this type of trajectory and show that it remains in other datasets. As z is independent, minimizing $\varepsilon$ amounts to $g(\theta, \cdot)$ learning $c(\cdot)$, regardless of the value of $p$ (Angluin & Laird, 1988). For more information about error and entropy relations in the presence of noisy labels see Appendix A.

---

[3]The minimal error is the error obtained by a maximum likelihood classifier that predicts y from ŷ.

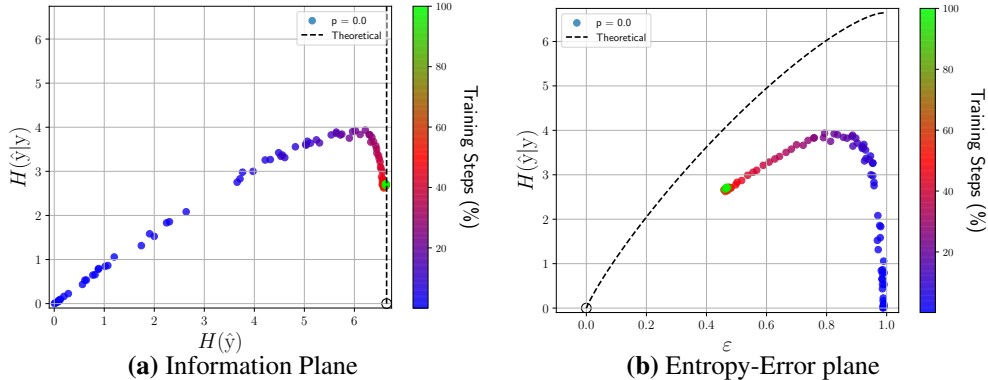

**Figure 2:** Learning trajectory of DenseNet on the CIFAR-100 dataset. The markers in the black dashed lines represent the ideal values of $H(\mathbf{y})$, $H(\hat{\mathbf{y}}|\mathbf{y})$ and $\varepsilon$ when $\tilde{\varepsilon} = 0$. **(a):** The dashed lines correspond to the maximum entropy value. **(b):** The dashed lines are the upper bound given by Fano's inequality.

## 3 CONNECTION BETWEEN SGD AND MARKOV CHAINS

In gradient based training of ANNs, the tunable parameters of the networks $\theta$ are changed in time by the gradient updates of the loss function, in order to minimize the learning error for a particular problem at hand. Let $\theta_n \in \Theta$ denote the tunable parameters of an ANN after $n \geq 1$ training steps of SGD. The parameters are initialized as $\theta_0$, which can be random or deterministic. The set $\Theta$ can be seen as a high dimensional Euclidean space with the network parameter $\theta$ as its vector.

At the training step $n$, the outcome of the learning algorithm is captured by the random variable $\hat{\mathbf{y}}_n$ which is modulated by the network function $g(\theta_n, \cdot)$ applied to the random input data $\mathbf{x}$, i.e., $\hat{\mathbf{y}}_n = g(\theta_n, \mathbf{x})$. Therefore the SGD learning gives rise the following sequence of random variables

$$g(\theta_0, \mathbf{x}), \ldots, g(\theta_n, \mathbf{x}), \ldots .$$

As $n$ grows large with a successful training, the sequence of random variables converges approximately to the true labels $\tilde{\mathbf{y}}$ which itself follows a joint distribution with $\mathbf{x}$. Note that the above random variables are coupled through the common random variable $\mathbf{x}$ and the sequence of parameter updates $\theta_n$.

If the probability distribution of $\hat{\mathbf{y}}_n$ is denoted by $\boldsymbol{p}_n$, a first question is to see how SGD methods modify $\boldsymbol{p}_n$ on the space of probability measures defined on $\mathbb{Y}$. As a consequence, one can determine the trajectory of $H(\hat{\mathbf{y}}_n)$, which will be plotted later. However it is additionally important in learning that the random variable $\hat{\mathbf{y}}_n$ approximates the true labels. Therefore, a second question would be how SGD methods change the joint distribution of $(\hat{\mathbf{y}}_n, \tilde{\mathbf{y}}_n)$. The answer could determine instead the trajectory of the conditional entropy $H(\hat{\mathbf{y}}_n|\tilde{\mathbf{y}}_n)$. We first study the trajectory of $H(\hat{\mathbf{y}}_n)$.

The random variables $\hat{\mathbf{y}}_n$ are defined as $g(\theta_n, \mathbf{x})$. Consider the sequence of random variables $\{\theta_n\}$. Let $\mathbb{T}$ denote the set of training samples $(\mathbf{x}, \mathbf{y})$ that are obtained prior to training and independently. In addition, let $\mathbb{T}_n$ be a subset of $\mathbb{T}$ that is used at the step $n$ for SGD update. $\mathbb{T}_n$ is assumed to be independent from $(\theta_0, \ldots, \theta_{n-1})$ and it is either deterministic and known all $n$ or randomly chosen at each step. These variations correspond to the variants of SGD.

In pure gradient based methods without momentum based techniques, the network parameters obey the following recursive relation

$$\theta_n = f(\theta_{n-1}, \mathbb{T}_n), \tag{2}$$

where $f$ denotes the update rule of SGD. The model assumes that the SGD updates only depend on the parameters in the last step and the training set used in the current iteration. We can assume that $\mathbb{T}_n$ are i.i.d. random variables if we neglect the effect of reusing training data in different batches. The first conclusion is that the sequence of random variables $\{\theta_n\}$ is a Markov chain.

**Proposition 2.** *The sequence of random variables $\{\theta_n\}$ defined as equation 2 with i.i.d. random variables $\mathbb{T}_n$ is a Markov Chain.*

*Proof.* If $\mathbb{T}_n$'s are i.i.d. random variables, the proof follows directly from Serfozo (2009, Proposition 11) on equation 2. □

The transition probability of this Markov chain can be obtained only from $f$ and $\mathbb{T}_1$. In that sense, the random process $\{\theta_n\}$ is a homogeneous Markov chain. Throughout this work, it is assumed that the Markov chain $\{\theta_n\}$ has a stationary distribution which corresponds to the learned ANN.

This proposition shows that SGD updates induce Markov property for weights of an ANN. The sequence of random variables $\hat{y}_n = g(\theta_n, \mathbf{x})$ however is in general not a Markov chain, particularly because they are coupled through a common random variable $\mathbf{x}$. Since we are interested in $H(\hat{y}_n)$ and the distribution $\boldsymbol{p}_n$, these random variables can be decoupled by considering the random variables $g(\theta_n, \mathbf{x}_n)$ where $\mathbf{x}_n$ are i.i.d. random variables with the same distribution as $\mathbf{x}$. Note that the value of the entropy function remains unchanged after decoupling, namely $H(\hat{y}_n) = H(g(\theta_n, \mathbf{x}_n))$. The new sequence is a function of a Markov chain and i.i.d. random variables. The question whether the resulting sequence is a Markov chain has been addressed in Spreij (2001); Gurvits & Ledoux (2005) showing that $\hat{y}_n$ is not a Markov chain in general unless certain conditions are met by the function $g(\cdot)$. Unfortunately the function $g(\cdot)$ is not injective and a non-injective function of a Markov chain is not Markov chain in general. However the random variable $g(\theta_n, \mathbf{x}_n)$ can be seen as the observation of the Markov process $\{\theta_n\}$ through a noisy memoryless channel $g(\cdot, \mathbf{x}_n)$. Therefore the random process $\{g(\theta_n, \mathbf{x}_n)\}$ is a HMP. See Ephraim & Merhav (2002) for an information theoretic survey.

If the learning is done perfectly, the HMP $\{g(\theta_n, \mathbf{x}_n)\}$ converges to the uniform distribution of correct labels. Since the random variables are discrete, entropy is a continuous function of the distribution $\boldsymbol{p}_n$. Therefore as the correct labels are uniformly distributed, the entropy $H(\hat{y}_n)$ approaches its maximum $\log K$. The instantaneous entropy $H(\hat{y}_n)$ would converge monotonically to the entropy of the stationary distribution $\log K$ if the sequence were to be a Markov chain. This could explain the monotonicity of $H(\hat{y}_n)$ in the experiments. Although the sequence not a Markov chain but it is indeed a HMP, the following proposition shows that the entropy is lower-bounded by an increasing function.

**Proposition 3.** *Suppose that the Markov process $\{\theta_n\}$ with the probability distribution $\boldsymbol{q}_n$ has a stationary distribution $\boldsymbol{q}$. We have*

$$H(\hat{y}_n) \geq \log |K| - D(\boldsymbol{q}_n \| \boldsymbol{q}).$$

The proof follows from data processing inequality for Kullback-Leibler divergence and is found in Appendix B. Note that since $\{\theta_n\}$ is a Markov process, $D(\boldsymbol{q}_n \| \boldsymbol{q})$ is non-increasing.

## 4 SGD on Joint Probability Measures

In the previous section, the non-decreasing property of $H(\hat{y}_n)$ was investigated by modeling the network output as an HMP. In the same spirit let us define $\tilde{y}_n = c(\mathbf{x}_n)$ to be i.i.d. realizations of $\tilde{y}$. In the ideal situation where the network manages to learn successfully the true labels, we can say $\hat{y}_n$ converges to $\tilde{y}_n$ almost surely[4]. The joint distribution of $(\hat{y}_n, \tilde{y}_n)$ specifies a point on the space of probability measures on $\mathbb{Y} \times \mathbb{Y}$. The task of learning consists tuning the parameters $\theta_n$ in a way that the joint distribution approaches the distribution of $(\tilde{y}_n, \tilde{y}_n)$. Therefore the gradient descent steps corresponds to a sequence of points, that is joint distributions of $(\hat{y}_n, \tilde{y}_n)$, on the space of probability measures on $\mathbb{Y} \times \mathbb{Y}$ with the end point ideally being the joint distribution of $(\tilde{y}_n, \tilde{y}_n)$. In this section, we investigate the gradient descent algorithm by exploring the path it takes on the space of probability measures on $\mathbb{Y} \times \mathbb{Y}$[5]. The trajectory of conditional entropy $H(\hat{y}_n | \tilde{y}_n)$ is determined for the trajectory of probability distributions on the space of joint measures.

However it is in general difficult to precisely characterize this path. Instead, one might ask how the gradient descent trajectory compares with a certain natural path on the space of distributions. A

---

[4]A weaker notion of convergence can be used although this choice is not crucial for the next results.

[5]The problem of finding the optimal way to change the distribution is connected to the problem of optimal transport. We would like to thank (removed for anonymity) for mentioning this connection.

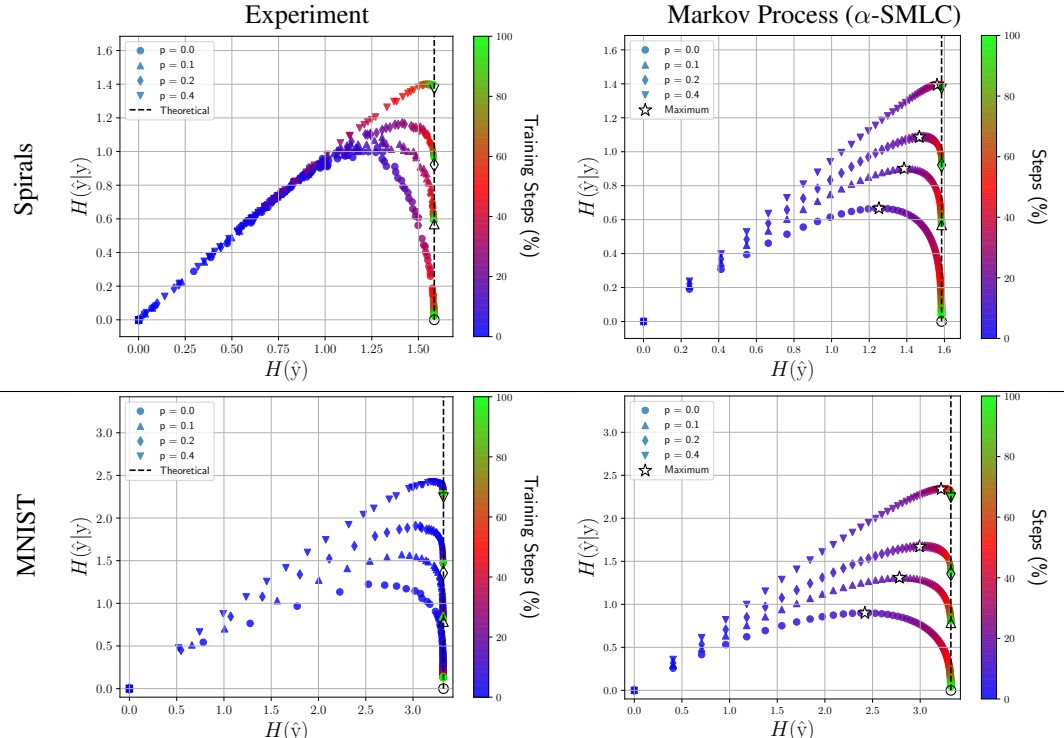

Figure 3: Experiments for FCNN and LeNet-5. This figures follow the same format as Figure 2(a). The shape of the makers differentiate between different values of $p$. For the $\alpha$-SMLC model the white star shows the inflection point of $H(\hat{\mathbf{y}}|\mathbf{y})$ and $\alpha = 0.85$.

relevant question is to ask what the shortest path between these probability measures is on this space and how similar is the trajectory of SGD compared to this shortest path. To be able to formally address this issue we require to define curves and lengths on the metric space of probability measures. The space of probability distributions defined on the discrete space $\mathbb{Y} \times \mathbb{Y}$, denoted by $\mathbb{P}(\mathbb{Y} \times \mathbb{Y})$, with total variation metric $d_{TV}(\cdot, \cdot)$ is a simplex in a finite dimensional Euclidean space. The total variation metric is equivalent to the $L^1$-distance between the points in the corresponding Euclidean space. A curve in this space is defined by a continuous function $\sigma : [0, 1] \to \mathbb{P}(\mathbb{Y} \times \mathbb{Y})$. The curve is called a shortest path if it has minimal length among all curves with endpoints $\sigma(0)$ and $\sigma(1)$. Note that the length is measured in this space using $L^1$-norm. The following theorem guarantees that there is a shortest path on this space between probability measures and it can be traveled using a Markov chain.

**Theorem 1.** *The shortest path between two probability measures $\mu$ and $\nu$ on the space of discrete probability measures $\mathbb{P}(\mathbb{Y} \times \mathbb{Y})$ is given by $t\mu + (1-t)\nu$ for $t \in [0, 1]$. Furthermore if the probability measures are represented by row vectors there is a transition matrix $\mathbf{\Pi}$ with the stationary distribution $\nu$ such that $\mu_n = \mu \mathbf{\Pi}^n$ is on the line segment between $\mu$ and $\nu$ and $\lim_{n \to \infty} \mu_n = \nu$.*

*Proof.* Not that the space of probability distributions here is a bounded compact metric space with each two points connected by a rectifiable curve. The existence of shortest path follows from (Burago et al., 2001, Corollary 2.5.20,Theorem 2.5.23). The transition matrix in Theorem 1 is given simply by

$$\mathbf{\Pi} = (1 - \alpha)\mathbf{I} + \alpha \mathbf{1}^T \nu$$

where $\mathbf{1} = (1, 1, \ldots, 1)$. $\qquad \square$

To see the implication of previous theorem more precisely, consider conditional distributions $P(\hat{\mathbf{y}}_n \in \cdot | \tilde{\mathbf{y}}_n = l)$ and let $\tilde{\boldsymbol{p}}_l(n) \in \mathbb{R}^K$ be the following vector for $l \in \mathbb{Y}$, that is

$$\tilde{\boldsymbol{p}}_l(n) = (P(\hat{\mathbf{y}}_n = 0 | \tilde{\mathbf{y}}_n = l), P(\hat{\mathbf{y}}_n = 1 | \tilde{\mathbf{y}}_n = l), \cdots, P(\hat{\mathbf{y}}_n = K - 1 | \tilde{\mathbf{y}}_n = l)) .$$

We use the compact notation $\tilde{\boldsymbol{P}}(n) \in \mathbb{R}^{K \times K}$ for the matrix with $\tilde{\boldsymbol{p}}_0(n), \ldots, \tilde{\boldsymbol{p}}_{K-1}(n)$ as rows. Note that, with the assumption that $\tilde{\mathbf{y}}_n$ is uniformly distributed, the joint distribution of $(\hat{\mathbf{y}}_n, \tilde{\mathbf{y}}_n)$ is fully determined by $\tilde{\boldsymbol{P}}(n)$ since $P(\hat{\mathbf{y}}, \tilde{\mathbf{y}}_n) = \frac{1}{K} P(\hat{\mathbf{y}}|\tilde{\mathbf{y}}_n)$. Suppose that the initialization $\theta_0$ is such that $g(\theta_0, \cdot)$ initially maps all inputs to the same class (the first class is assumed for simplicity). Therefore the initial distribution matrix $\tilde{\boldsymbol{P}}(0)$ assumes no knowledge about the input and is given by

$$\tilde{\boldsymbol{P}}(0) = \mathbf{1}\boldsymbol{e}_1^{\mathrm{T}} .$$

Ideally, in a learning algorithm, the matrix $\tilde{\boldsymbol{P}}(0)$ converges to an optimal distribution $\tilde{\boldsymbol{P}}^*$ as $n \to \infty$, that is $\tilde{\boldsymbol{P}}^* = \lim_{n\to\infty} \tilde{\boldsymbol{P}}(n)$. If $P(\hat{\mathbf{y}}_n = \tilde{\mathbf{y}}_n) = 1$, we have

$$\tilde{\boldsymbol{P}}^* = \boldsymbol{I} .$$

Now that we set the initial distribution and stationary distributions, the following transition matrix provides a way to pass from the initial distribution to the stationary one.

**Definition 1.** $\alpha$- *Simple Markov Learning Chain ($\alpha$-SMLC)*
*Given $0 < \alpha < 1$, the sequence of random pairs $\{(\hat{\mathbf{y}}_n, \tilde{\mathbf{y}}_n)\}$ is an $\alpha$-SMLC if $\tilde{\boldsymbol{p}}_l(n) = \tilde{\boldsymbol{p}}_l(n-1)\boldsymbol{\Pi}_l$, for every $n \geq 0$, $l \in \mathbb{Y}$ with*

$$\boldsymbol{\Pi}_l = \mathbf{1}\boldsymbol{e}_l^{\mathrm{T}} + \alpha(\boldsymbol{I} - \mathbf{1}\boldsymbol{e}_l^{\mathrm{T}}),$$
$$\tilde{\boldsymbol{P}}(0) = \mathbf{1}\boldsymbol{e}_l^{\mathrm{T}} .$$

The above construction provides a different transition matrix for each $\tilde{\boldsymbol{p}}_l(n)$ depending on $l$. The following theorem describes how an $\alpha$-SMLC moves $\tilde{\boldsymbol{P}}(n)$ in the space of stochastic matrices. It actually shows this construction leads to points on the shortest path between the measures.

**Theorem 2.** *If $(\hat{\mathbf{y}}_n, \tilde{\mathbf{y}}_n)$ is the $n$-th random pair generated of an $\alpha$-SMLC, then*

$$\tilde{\boldsymbol{P}}(n) = (1 - t)\tilde{\boldsymbol{P}}(0) + t\boldsymbol{I} , \ \ with \ t = (1 - \alpha^n). \tag{3}$$

*Proof.* c.f. Appendix B $\qquad\qquad\qquad\qquad\qquad\qquad\qquad\qquad\qquad\qquad\qquad\qquad\qquad\square$

This shows that $\tilde{\boldsymbol{P}}(n)$ belongs to the continuous curve from equation 3, regardless of the choice of $\alpha$. Moreover, let $\hat{\mathbf{y}}(t)$ be a version of $\hat{\mathbf{y}}_n$ parametrized by $t \in [0, 1]$ such that its conditional distribution with $\tilde{\mathbf{y}}_n$ corresponds to $\tilde{\boldsymbol{P}}(n) = (1 - t)\tilde{\boldsymbol{P}}(0) + t\boldsymbol{I}$.

**Proposition 4.** *If $\{(\hat{\mathbf{y}}_n, \tilde{\mathbf{y}}_n)\}$ is an $\alpha$-SMLC and $\mathbf{z}$ follows the distribution*

$$P(\mathbf{z} = i) = \begin{cases} 1 - p, & i = 0 \\ \frac{p}{K-1}, & i \in \{1, \ldots, K - 1\} \end{cases} ,$$

*then*

$$\frac{\partial H(\hat{\mathbf{y}}(t)|\mathbf{y})}{\partial t} = \frac{1}{K} \Big[ p \log(1 - tp) + (1 - K + p) \log(1 - t + t\frac{p}{K - 1}) \\ - (K - 1)(1 - p) \log(t(1 - p)) - (K - 1)p \log(t\frac{p}{K - 1}) \Big]$$

*Proof.* c.f. Appendix B $\qquad\qquad\qquad\qquad\qquad\qquad\qquad\qquad\qquad\qquad\qquad\qquad\qquad\square$

**Corollary 1.** *In the setting of Proposition 4, if $p \to 0$ then $H(\hat{\mathbf{y}}(t)|\mathbf{y})$ has one maximum at $t = \frac{1}{2}$, $H(\hat{\mathbf{y}}(\frac{1}{2})|\mathbf{y}) = \frac{K-1}{K} \log 2$.*

This result allows us to characterize the shape of the 2D curves $(H(\hat{\mathbf{y}}(t)), H(\hat{\mathbf{y}}(t)|\mathbf{y}))$ for the above construction.

We now consider the implication of Markov assumption for error. Define $\hat{\mathbf{y}}_n^l \in \mathbb{Y}$ for all $l \in \mathbb{Y}$ to be random variables distributed according to the conditional probabilities $P(\hat{\mathbf{y}}_n^l = k) = P(g(\theta_n, \mathbf{x}) = k|\tilde{\mathbf{y}}_n = l)$ for all $k \in \mathbb{Y}$. Note that, since $\tilde{\mathbf{y}}_n$ is uniformly distributed, one can compute the joint distribution of $(\hat{\mathbf{y}}_n, \tilde{\mathbf{y}}_n)$ from the marginal distributions of $\hat{\mathbf{y}}_n^0, \ldots, \hat{\mathbf{y}}_n^{K-1}$ and vice-versa. The following propositions shows that the Markov trajectory implies the reduction in error as well.

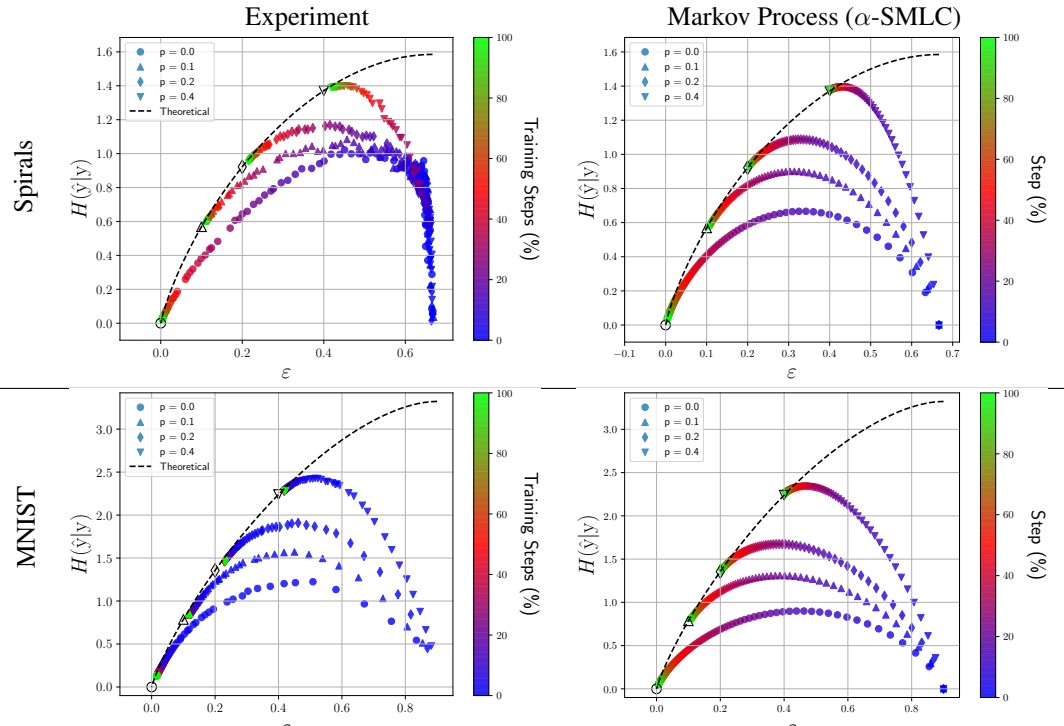

Figure 4: Experiments for FCNN and LeNet-5. This figures follow the same format as Figure 2(b), with $\alpha = 0.85$. The shape of the makers differentiate between different values of $p$.

**Proposition 5.** *If $\{\hat{y}_n^l\}$ is a stationary Markov Chain converging to the distribution $\tilde{p}^* = e_l$ then*

$$P(\hat{y}_n = l|\tilde{y}_n = l) \leq P(\hat{y}_{n+1} = l|\tilde{y}_n = l)$$

**Corollary 2.** *If $\{\hat{y}_n^l\}$ is a stationary Markov Chain converging to the distribution $\tilde{p}^* = e_l$ for all $l \in \mathbb{Y}$ then*

$$P(\hat{y}_n \neq \tilde{y}_n) \geq P(\hat{y}_{n+1} \neq \tilde{y}_n).$$

This last results shows that $\tilde{\varepsilon}$ is non-increasing with $n$ (i.e., no over-fitting), which is a desirable property for any learning algorithm. We will show through numerical simulations how a comparable behavior is observed for gradient descent methods.

## 5 EXPERIMENTS

In this section we compare gradient based learning to the $\alpha$-SMLC model through empirical simulations. First, we use the datasets where SGD is extremely successful at the classification task ($\tilde{\varepsilon} \approx 0$), that is the MNIST dataset and the spirals dataset (Anonymous, 2018). The spirals dataset constitutes a 2D-spiral classification task constructed as

$$\mathbf{x} = \left( \left(\sqrt{a} + b\right) \cos\left(2\pi a + \tfrac{2\pi}{K}\tilde{y}\right), \quad \left(\sqrt{a} + b\right) \sin\left(2\pi a + \tfrac{2\pi}{K}\tilde{y}\right) \right),$$

where $a \in [0, 1]$, $b \in [0, 0.1]$ and $\tilde{y} \in \{0, 1, \dots, K - 1\}$ are independent uniformly distributed random variables and $K = 3$. This dataset is divided into a training set of $50\,000$ samples and a test set of $2\,000$. Furthermore, we train the FCNN of Anonymous (2018) for the spirals dataset and use LetNet-5 (LeCun et al., 1999) for MNIST, achieving an average accuracy above $99\%$ in both cases. For the sake of completeness we perform more experiments on the CIFAR-10 and CIFAR-100 datasets using the DenseNet architecture from Huang et al. (2017) with 40 and 100 layers respectively. FCNN and LetNet-5 are trained using Adam's optimizer, while DenseNet is trained with SGD. More detailed explanation about the experimental setup is provided in Appendix C. We estimate use a naive estimator of entropy which consists on computing the empirical distribution of

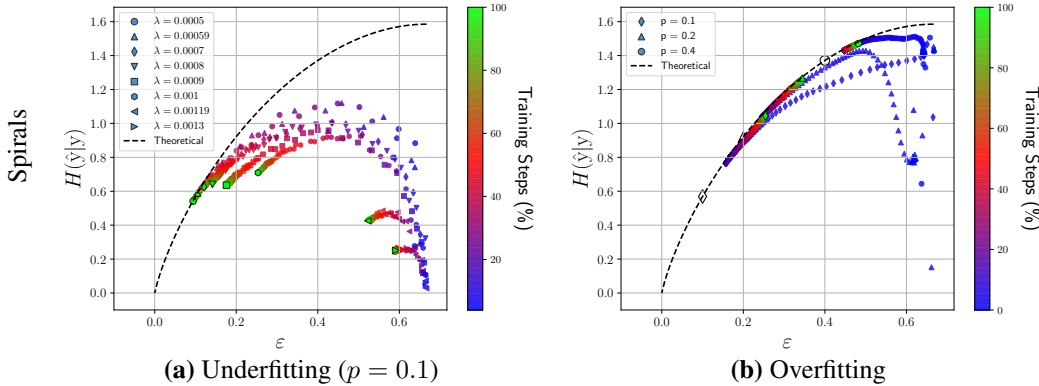

**(a)** Underfitting ($p = 0.1$)  **(b)** Overfitting

Figure 5: Experiments for FCNN on the spirals dataset. This figures follow the same format as Figures 3 and 4.

$(\hat{y}, y)$ and directly calculating entropy afterwards. This method is known to have an approximation error of $K^2/N$ (Miller, 1955), which is good enough for our experiments since $N$ is much larger than $K^2$ in our datasets. For cases with larger $K$ one could use more sophisticated methods, such as Schürmann (2004); Archer et al. (2014). For our experiments we introduce i.i.d. noise to the dataset labels before training according to $P(z = 0) = 1 - p$ and $P(z \neq 0) = p/(K - 1)$. We keep fixed $\alpha = 0.85$ for all simulations.

In Figure 3 we show the similarity between the $\alpha$-SMLC model and ANNs on the spirals and MNIST dataset. This figure is obtained by averaging the entropy values over 100 realizations of training. We observe ANNs move along the information plane in a similar way as the $\alpha$-SMLC model, and converge to the optimal distribution. In addition, we display the inflection point of $H(y|\hat{y})$ of the $\alpha$-SMLC model for different values of $p$. Interestingly, as labels get noisier the inflection point occurs at a larger $H(\hat{y})$ value. This seems to be the case for SGD learning as well. This phenomena suggests that, as labels get noisier, a learning algorithm needs to know more about the input distribution before it can start assigning labels efficiently. Similar conclusions can be drawn form Figure 4, which is also obtained by averaging over 100 realizations of training. An interesting result is that, regardless of the value of $p$, a good learning algorithm should converge to a point that lies on the upper bound provided by Fano's inequality. These Figures artificially mitigate the randomness induced by SGD on the trajectory by averaging over several realizations of training [6]. How much SGD oscillates seems to depend on the experimental setup, such as the learning rate, dataset, and the structure of the classifier. See Appendix C for examples of highly oscillating trajectories.

Our last experiment consists on investigating how underfitting and overfitting affects the trajectory of SGD. To that end in Figure 5(a) we include an $\ell_1$ norm regularization term into the loss function (details in Appendix C) controlled by a parameter $\lambda$. As expected for sufficiently small $\lambda$ the minimum error is attained. Interestingly, as we induce underfitting by increasing this regularization coefficient the obtained models move away from Fano's bound. This naturally leads to increased error values. On the other hand, in Figure 5(b) we increase the number of parameters of FCNN and reduce the dataset size in order to induce overfitting (details in Appendix C). While overfitting leads to larger error as well as underfitting, it can be distinguished by its trajectory in the entropy-error plane. Underfitting seems to push models away from Fano's bound while overfitting happens when an ANN is at the bound. This experiment shows that the information plane provides a richer view beyond train and test error that allow us to observe effects that were previously hidden. Further understanding about desired trajectories is interesting since it may allow practitioners to monitor models during training, spot undesired behaviors, and possibly tune hyperparameters accordingly.

---

[6] The realizations that did not converge are discarded.

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

## A    COLLECTION OF RESULTS FOR LEARNING WITH RANDOM LABELS

For $\varepsilon < 1 - \frac{1}{K}$, proposition 1 can be written as a lower bound on $\varepsilon$ through the following Proposition of Anonymous (2018).

**Proposition 6.** *(Anonymous, 2018)* $\Phi(H(\mathrm{z})) \leq \varepsilon$, *where* $\Phi : [0, \log|K|] \to [0, 1 - \frac{1}{K}]$ *is the inverse function of* $\Psi$ *in the interval* $[0, 1 - \frac{1}{K}]$.

*Proof.* For independent noise z, we have

$$H(\mathrm{y}|\hat{\mathrm{y}}) \geq H(\mathrm{y}|\hat{\mathrm{y}}, \tilde{\mathrm{y}}) = H(\mathrm{z}|\hat{\mathrm{y}}, \tilde{\mathrm{y}}) = H(\mathrm{z}).$$

By Proposition 1, it follows that $H(\mathrm{z}) \leq \Psi(\varepsilon)$. Applying the inverse $\Phi(x), x \in [0, 1 - \frac{1}{K}]$, completes the proof.    □

In Anonymous (2018) it is shown that this bound is sharp when z is distributed such that $\Phi(H(\mathrm{z})) = p$, thus $p \leq \varepsilon$. We generalize this result in the following theorem for an arbitrary distribution of z, under some mild conditions, and show that $p \leq \varepsilon$ is in fact a sharp lower bound for arbitrary z.

**Lemma 1.** *Let* z $\in \mathbb{Y}$ *be a random variable with* $P(\mathrm{z} = 0) = 1 - p$, *then*

$$H(\mathrm{z}) \leq \Psi(p).$$

*Proof.* Let us define $\beta_k \triangleq P(\mathrm{z} = k)$ and the auxiliary random variable $\tilde{\mathrm{z}} \in \{1, \ldots, K-1\}$ with $P(\tilde{\mathrm{z}} = k) = \beta_k / p$ for all $k = 1, \ldots, K - 1$.

$$
\begin{aligned}
H(\mathrm{z}) &= -(1-p)\log(1-p) - \sum_{k=1}^{K-1} \beta_k \log \beta_k \\
&= -(1-p)\log(1-p) - p\sum_{k=1}^{K-1} \frac{\beta_k}{p} \log \frac{\beta_k}{p} p \\
&= -(1-p)\log(1-p) - p\sum_{k=1}^{K-1} \frac{\beta_k}{p} \log p - p\sum_{k=1}^{K-1} \frac{\beta_k}{p} \log \frac{\beta_k}{p} \\
&= -(1-p)\log(1-p) - \sum_{k=1}^{K-1} \beta_k \log p + pH(\tilde{\mathrm{z}}) \\
&= -(1-p)\log(1-p) - p\log p + pH(\tilde{\mathrm{z}}) \\
&= h_b(p) + pH(\tilde{\mathrm{z}}) \\
&\leq h_b(p) + p\log(K-1) \\
&= \Psi(p)
\end{aligned}
$$

□

**Theorem 3.** *If* $P(\mathrm{z} = 0) = 1 - p$ *and* $P(\mathrm{z} = 0) > P(\mathrm{z} = k)$ *for all* $k = 1, \ldots, K - 1$, *then*

$$p \leq \varepsilon,$$

*and equality is attained if and only if* $P(\hat{\mathrm{y}} = \tilde{\mathrm{y}}) = 1$.

*Proof.* of Theorem 3 For the sake of notation let us define

$$
\begin{aligned}
\delta_k &\triangleq P(\hat{\mathrm{y}} = \tilde{\mathrm{y}} \oplus k) \\
\beta_k &\triangleq P(\mathrm{z} = k) \text{ for all } k \in \{0, \ldots, K - 1\} \\
\beta_{\max} &\triangleq \max_{k=1,\ldots,K-1} \beta_k .
\end{aligned}
$$

From Proposition 1 we know that $\varepsilon \geq \Phi(H(\mathrm{z}))$. Since $\Psi$ is an increasing function in the interval $[0, 1 - \frac{1}{K}]$ and $\Phi$ is its inverse in that interval, we have that $\Phi$ is an increasing function as well. From Lemma 1 (c.f. Appendix B) we know that $H(\mathrm{z}) \leq \Psi(p)$, this leads to

$$\Phi(H(\mathrm{z})) \leq \Phi(\Psi(p)) = p.$$

Then, if the bound from Proposition 1 were to be sharp, $\varepsilon$ could reach values strictly lower than $p$. We will show that this is not possible.

$$1 - \varepsilon = P(\hat{\mathbf{y}} = \mathbf{y}) \tag{4}$$

$$= \sum_{k=0}^{K-1} P(\hat{\mathbf{y}} = \mathbf{y}|\mathbf{z} = k)P(\mathbf{z} = k) \tag{5}$$

$$= \sum_{k=0}^{K-1} P(\hat{\mathbf{y}} = \mathbf{y} \oplus k)P(\mathbf{z} = k) \tag{6}$$

$$= \sum_{k=0}^{K-1} \delta_k \beta_k \tag{7}$$

$$= (1-p)\delta_0 + \sum_{k=1}^{K-1} \beta_k \delta_k \tag{8}$$

$$\leq (1-p)\delta_0 + \sum_{k=1}^{K-1} \beta_{\max}\delta_k \tag{9}$$

$$= (1-p)\delta_0 + \beta_{\max}(1-\delta_0) \tag{10}$$

$$\leq (1-p)\delta_0 + (1-p)(1-\delta_0) \tag{11}$$

$$= (1-p). \tag{12}$$

If $\varepsilon < p$ we obtain $(1-p) < 1 - \varepsilon \leq (1-p)$ which is a contradiction, hence it must hold that $\varepsilon \geq p$.

Finally, if $\varepsilon = p$ then equation 10 yields

$$1 - p \leq (1-p)\delta_0 + \beta_{\max}(1-\delta_0).$$

Since $\beta_{\max} < (1-p)$, this inequality holds if and only if $\delta_0 = 1$, that is $P(\hat{\mathbf{y}} = \tilde{\mathbf{y}}) = 1$. $\qquad\square$

This theorem shows that the minimum expected error $\varepsilon$ can only be attained if $g(\theta, \cdot)$ manages to denoise the labels, hence $\tilde{\varepsilon} = 0$. We extend this result by deriving bounds for $\tilde{\varepsilon}$, given $\varepsilon$ and $p$, in the following theorem.

**Theorem 4.** *(Angluin & Laird, 1988)*
*Given $\varepsilon < 1 - \frac{1}{K}$ and $p < \frac{1}{2}$, if $P(\mathbf{z} = 0) = 1 - p$, and $P(\mathbf{z} = k) < \frac{1}{2}$ for all $k = 1, \ldots, K-1$, then $\tilde{\varepsilon}$ is bounded by*

$$\frac{\varepsilon - p}{1-p} \leq \tilde{\varepsilon} \leq \frac{\varepsilon - p}{1 - 2p}.$$

*Proof.* of Theorem 4

$$1 - \varepsilon = P(\hat{\mathbf{y}} = \mathbf{y})$$

$$= \sum_{k=0}^{K-1} \underbrace{P(\tilde{\mathbf{y}} \oplus k = \hat{\mathbf{y}})}_{\delta_k} \underbrace{P(\mathbf{z} = k)}_{\beta_k}$$

$$= (1-p)(1-\tilde{\varepsilon}) + \sum_{k=1}^{K-1} \delta_k \beta_k \tag{13}$$

$$\leq (1-p)(1-\tilde{\varepsilon}) + \beta_{\max} \sum_{k=1}^{K-1} \delta_k$$

$$= (1-p)(1-\tilde{\varepsilon}) + \beta_{\max}\tilde{\varepsilon}$$

$$= (1-p) - \tilde{\varepsilon}((1-p) - \beta_{\max}), \tag{14}$$

$$\tilde{\varepsilon} \leq \frac{(1-p)-(1-\varepsilon)}{(1-p)-\beta_{\max}} \leq \frac{(1-p)-(1-\varepsilon)}{(1-p)-p} = \frac{\varepsilon-p}{1-2p}.$$

Applying $\sum_{k=1}^{K-1} \delta_k \beta_k \geq 0$ in equation 13 leads to $1 - \varepsilon \geq (1-p)(1-\tilde{\varepsilon})$ thus

$$\frac{(1-p)-(1-\varepsilon)}{(1-p)} \leq \tilde{\varepsilon} \quad \Rightarrow \quad \frac{\varepsilon-p}{1-p} \leq \tilde{\varepsilon},$$

which completes the proof. $\qquad\square$

**Corollary 3.** *If $p < \frac{1}{2}$, $\varepsilon < 1 - \frac{1}{K}$, and $\frac{\varepsilon-p}{1-2p} < 1 - \frac{1}{K}$ then*

$$\max\{H(\tilde{\mathbf{y}}|\hat{\mathbf{y}}), H(\hat{\mathbf{y}}|\tilde{\mathbf{y}})\} \leq \Psi(\frac{\varepsilon-p}{1-2p}).$$

*Proof.* Since $\Psi$ is an increasing function in the interval $[0, 1 - \frac{1}{K}]$, the proof follows from applying Theorem 4 on Proposition 1. $\qquad\square$

In information theory there is a result of this kind, known as Mrs. Gerber's Lemma (MGL), that does not require knowledge about $p$ and $\varepsilon$. MGL provides an upper bound on $H(\tilde{\mathbf{y}}|\hat{\mathbf{y}})$ given $H(\mathbf{y}|\hat{\mathbf{y}})$ for the case of $K = 2$. This result also states that the minimum $H(\mathbf{y}|\hat{\mathbf{y}})$ is attained when $H(\tilde{\mathbf{y}}|\hat{\mathbf{y}}) = 0$. Since we assumed $\tilde{\mathbf{y}}$ to be uniformly distributed, this corresponds to $\tilde{\varepsilon} = 0$, up to permutation ambiguities. Generalizing MGL for arbitrary $K$ is still an open question in information theory. Nevertheless, Jog & Anantharam (2014) successfully proved MGL for the cases where $K$ is a power of 2. We summarize that result in the following proposition.

**Proposition 7.** *Generalized Mrs. Gerber's Lemma for $K = 2^n$ (Jog & Anantharam, 2014)*

$$f_{2^n}(\tilde{y}, z) = \min_{H(\tilde{\mathbf{y}}|\hat{\mathbf{y}})=\tilde{y}, H(z)=z} H(\tilde{\mathbf{y}} \oplus \mathbf{z}|\hat{\mathbf{y}})$$

*with*

$$f_{2^n}(\tilde{y}, z) = \begin{cases} f_2(\tilde{y} - k\log 2, z - k\log 2) + k\log 2 & \text{if } k\log 2 \leq \tilde{y}, z \leq (k+1)\log 2, \\ \max(\tilde{y}, z) & \text{otherwise} \end{cases}$$

*where*

$$f_2(x, y) = h(h^{-1}(x) \star h^{-1}(y))$$

*and $k$ is an arbitrary positive integer and $a \star b \triangleq a(1-b) + b(1-a)$.*

We have derived inequalities that relate entropies and error values. Then we showed that in the presence of corrupted labels, the best $g(\theta, \cdot)$ can do for minimizing $\varepsilon$ is to learn $c$, regardless of the value of $p$.

# B    DEFERRED PROOFS

Proof of Proposition 3: The proof follows from the following general theorem.

**Theorem 5.** *Let $\{\mathbf{y}_n\}$ be an HMP defined as the observation of a Markov process $\{\mathbf{x}_n\}$ through an arbitrary stationary memoryless channel with values in the state space $\mathbb{Y}$. Suppose that the probability distributions on the respective state spaces of $\{\mathbf{x}_n\}$ and $\{\mathbf{y}_n\}$ are given by $\{\boldsymbol{q}_n\}$ and $\{\boldsymbol{p}_n\}$ with the stationary distribution $\boldsymbol{q}$ and $\boldsymbol{p}$. Then*

$$D(\boldsymbol{p}_n\|\boldsymbol{p}) \leq \log D(\boldsymbol{q}_n\|\boldsymbol{q}).$$

*Proof.* Based on the assumption above, $\mathbf{x}_n$ and $\mathbf{y}_n$ are related according to the conditional distribution characterized by the conditional probabilities $\{\boldsymbol{r}(\cdot|x) : x \in \mathbb{X}\}$. Denote the joint distribution of $\mathbf{x}_n$ and $\mathbf{y}_n$ by $\boldsymbol{\mu}_n$ defined on $\mathbb{X} \times \mathbb{Y}$ and given by $\boldsymbol{q}_n(x) \times \boldsymbol{r}(\cdot|x)$. The stationary joint distribution $\boldsymbol{\mu}$ is defined by $\boldsymbol{q} \times \boldsymbol{r}$. Using the chain rule (Cover & Thomas, 2006, Theorem 2.5.3), we have:

$$D(\boldsymbol{\mu}_n\|\boldsymbol{\mu}) = D(\boldsymbol{q}_n\|\boldsymbol{q}) + D(\boldsymbol{r}\|\boldsymbol{r}) = D(\boldsymbol{q}_n\|\boldsymbol{q}).$$

On the other hand, the chain rule and the non-negativity of Kullback-Leibler divergence shows that:

$$D(\boldsymbol{\mu}_n\|\boldsymbol{\mu}) \geq D(\boldsymbol{p}_n\|\boldsymbol{p}),$$

which implies the theorem. $\qquad\square$

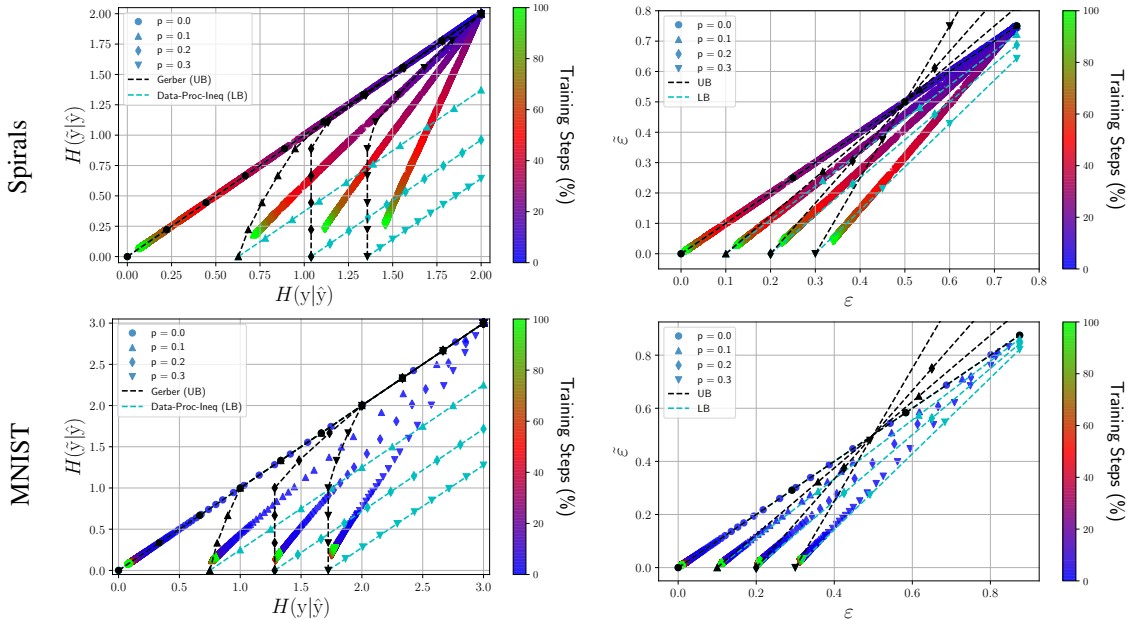

Figure 6: Mrs Gerber's Lemma in action.

Using the fact that the stationary distribution of $\hat{y}_n$ is equal to the uniform distribution, we have:

$$D(\hat{y}_n \| \hat{y}) = \log |K| - H(\hat{y}_n).$$

This fact along with Theorem 5 proves Proposition 3.

*Proof.* of Theorem 2
Since $\mathbf{1}e_l^T \mathbf{1}e_l^T = \mathbf{1}e_l^T$ we get

$$(\mathbf{\Pi}_l)^2 = \underbrace{\mathbf{1}e_l^T \mathbf{1}e_l^T}_{\mathbf{1}e_l^T} + \alpha \underbrace{\mathbf{1}e_l^T(\boldsymbol{I} - \mathbf{1}e_l^T)}_{\mathbf{0}} + \alpha \underbrace{(\boldsymbol{I} - \mathbf{1}e_l^T)\mathbf{1}e_l^T}_{\mathbf{0}} + \alpha^2(\boldsymbol{I} - \mathbf{1}e_l^T)$$

$$= \mathbf{1}e_l^T + \alpha^2(\boldsymbol{I} - \mathbf{1}e_l^T).$$

By induction we get

$$(\mathbf{\Pi}_l)^n = \mathbf{1}e_l^T + \alpha^n(\boldsymbol{I} - \mathbf{1}e_l^T) = (1 - \alpha^n)\mathbf{1}e_l^T + \alpha^n \boldsymbol{I}.$$

Replacing $t = (1 - \alpha^n) \in [0, 1]$ yields

$$(\mathbf{\Pi}_l)^n = t\mathbf{1}e_l^T + (1 - t)\boldsymbol{I}.$$

Finally,

$$\begin{aligned}
\tilde{\boldsymbol{p}}_l(n) &= \tilde{\boldsymbol{p}}_l(0)(\mathbf{\Pi}_l)^n \\
&= \tilde{\boldsymbol{p}}_l(0)(t\mathbf{1}e_l^T + (1 - t)\boldsymbol{I}) \\
&= (1 - t)\tilde{\boldsymbol{p}}_l(0) + te_l,
\end{aligned}$$

thus

$$\tilde{\boldsymbol{P}}_l(n) = (\tilde{\boldsymbol{p}}_0(n)^T, \dots, \tilde{\boldsymbol{p}}_{K-1}(n)^T)^T = (1 - t)\tilde{\boldsymbol{P}}_l(0) + \tilde{t}\boldsymbol{I}.$$

$\square$

*Proof.* of Proposition 4
Since $y = \tilde{y} \oplus z$ the distribution of y can be expressed as a circular convolution between the distributions of z and $\tilde{y}$. Informally, we express this as $P(y) = P(\tilde{y} \oplus z) = P(\tilde{y}) \circledast P(z)$. The same holds true if we condition by $\hat{y}$, that is $P(y|\hat{y} = l) = P(\tilde{y} \oplus z|\hat{y} = l) = P(\tilde{y}|\hat{y} = l) \circledast P(z)$ for all

$l = 0, \ldots, K - 1$. Then for uniformly distributed y, and z distributed according to equation 4, we can express this relation in matrix form as

$$\boldsymbol{P}(t) = \underbrace{\left((1 - p - \frac{p}{K-1})\boldsymbol{I} + \frac{p}{K-1}\boldsymbol{1}\boldsymbol{1}^{\mathrm{T}}\right)}_{\text{circular convolution matrix of } P(z)} \tilde{\boldsymbol{P}}(t).$$

Therefore, the $(i, j)$-th element of $\boldsymbol{P}(t)$ is denoted as $P_{i,j}(t)$ and given by

$$P_{i,j}(t) = \boldsymbol{e}_i^{\mathrm{T}} \boldsymbol{P}(t) \boldsymbol{e}_j = (1 - t)\boldsymbol{e}_1^{\mathrm{T}} \boldsymbol{e}_j + t(1 - p - \frac{p}{K-1})\boldsymbol{e}_i^{\mathrm{T}} \boldsymbol{e}_j + t\frac{p}{K-1}.$$

Differentiating $P_{i,j}(t)$ with respect to $t$ yields

$$\frac{\partial P_{i,j}(t)}{\partial t} = -\boldsymbol{e}_1^{\mathrm{T}} \boldsymbol{e}_j + (1 - p - \frac{p}{K-1})\boldsymbol{e}_i^{\mathrm{T}} \boldsymbol{e}_j + \frac{p}{K-1}.$$

For particular choices of $i, j$ these expressions boil down to

$$P_{i,j}(t) = \begin{cases} 1 - t + t(1 - p), & j = 1, i = j \\ 1 - t + t\frac{p}{K-1}, & j = 1, i \neq j \\ t(1 - p), & j \neq 1, i = j \\ t\frac{p}{K-1}, & j \neq 1, i \neq j \end{cases} \quad \text{and} \quad \frac{\partial P_{i,j}(t)}{\partial t} = \begin{cases} -p, & j = 1, i = j \\ \frac{p}{K-1} - 1, & j = 1, i \neq j \\ 1 - p, & j \neq 1, i = j \\ \frac{p}{K-1}, & j \neq 1, i \neq j \end{cases}.$$

Finally, we make use of these to derive a closed form expression for $\frac{H(\hat{\mathbf{y}}(t)|\mathbf{y})}{\partial t}$, that is

$$\frac{H(\hat{\mathbf{y}}(t)|\mathbf{y})}{\partial t} = \frac{\partial}{\partial t}\left[-\sum_{j=1}^{K} \frac{1}{K} \sum_{i=1}^{K} P_{i,j}(t) \log P_{i,j}(t)\right]$$

$$= -\frac{1}{K}\sum_{j=1}^{K}\sum_{i=1}^{K}\left[\frac{\partial P_{i,j}(t)}{\partial t}(1 + \log P_{i,j}(t))\right]$$

$$= -\frac{1}{K}\sum_{j=1}^{1}\sum_{i=j}\left[\frac{\partial P_{i,j}(t)}{\partial t}(1 + \log P_{i,j}(t))\right] - \frac{1}{K}\sum_{j=1}^{1}\sum_{i \neq j}\left[\frac{\partial P_{i,j}(t)}{\partial t}(1 + \log P_{i,j}(t))\right]$$

$$- \frac{1}{K}\sum_{j=2}^{K}\sum_{i=j}\left[\frac{\partial P_{i,j}(t)}{\partial t}(1 + \log P_{i,j}(t))\right] - \frac{1}{K}\sum_{j=2}^{K}\sum_{i \neq j}\left[\frac{\partial P_{i,j}(t)}{\partial t}(1 + \log P_{i,j}(t))\right]$$

$$= \frac{1}{K}p\log(1 - tp) + \frac{1}{K}(K - 1)(\frac{p}{K-1} - 1)(1 + \log(1 - t + t\frac{p}{K-1}))$$

$$- \frac{1}{K}(K - 1)(1 - p)(1 + \log(t(1 - p))) - \frac{1}{K}(K - 1)p(1 + \log(t\frac{p}{K-1}))$$

$$\frac{1}{K}\underbrace{\left(p - (K - 1)(\frac{p}{K-1} - 1) - (K - 1)(1 - p) - (K - 1)p\right)}_{=0}$$

$$= \frac{1}{K}\left[p\log(1 - tp) + (1 - K + p)\log(1 - t + t\frac{p}{K-1})\right.$$

$$\left. - (K - 1)(1 - p)\log(t(1 - p)) - (K - 1)p\log(t\frac{p}{K-1})\right].$$

**Remark**: For $p \to 0$ we now that the maximum the curve is at $t = \frac{1}{2}$ since

$$\lim_{p \to 0} \frac{H(\hat{\mathbf{y}}(t)|\mathbf{y})}{\partial t} \overset{!}{=} 0 \Rightarrow \frac{1}{K}[(1 - K)\log(1 - t) - (1 - K)\log t] = 0 \Rightarrow \log\frac{1 - t}{t} = 0 \Rightarrow t = \frac{1}{2}.$$

$\square$

Table 1: Simulation Parameters for Figure 7 (Anonymous, 2018). The models with highest accuracy are used in the main paper.

| Dataset | Activation | Batch Size | $\gamma_{\max}$ | $\gamma_{\min}$ | Test Accuracy |
|---------|-----------|-----------|-----------------|-----------------|---------------|
| Spirals | tanh | 128 | $10^{-1}$ | $10^{-2}$ | 99.7% |
| | sigmoid | 128 | $10^{-1}$ | $10^{-5}$ | 99.6% |
| | ReLU | 700 | $10^{-1}$ | $10^{-5}$ | 97.8% |
| MNIST | tanh | 128 | $10^{-2}$ | $10^{-2}$ | 97.1% |
| | sigmoid | 128 | $10^{-2}$ | $10^{-4}$ | 96.3% |
| | ReLU | 128 | $10^{-2}$ | $10^{-4}$ | 99.1% |

Table 2: Simulation Parameters for DenseNet on CIFAR datasets

| Dataset | Activation | Batch Size | $\gamma_{\max}$ | $\gamma_{\min}$ | Test Accuracy |
|---------|-----------|-----------|-----------------|-----------------|---------------|
| CIFAR-10 | ReLU | 64 | $10^{-1}$ | $10^{-1}$ | 80.2% |
| CIFAR-100 | ReLU | 64 | $10^{-1}$ | $10^{-1}$ | 80.2% |

## C  EXPERIMENTAL DETAILS AND COMPLEMENTARY EXPERIMENTS

A fully connected ANN with four hidden layers of five neurons each, as FCNN, is trained on the spirals dataset. For the MNIST dataset, the popular convolutional network called LeNet-5 LeCun et al. (1999) is used. To train these networks we let the learning rate $\gamma \in \mathbb{R}$ start at a given $\gamma_{\max} \in \mathbb{R}$ and then decay by 40% per epoch until reaching some given minimum learning rate $\gamma_{\min} < \gamma_{\max}$, that is $\gamma = \max\{\gamma_{\max} 0.6^{\lfloor \text{epoch} \rfloor}, \gamma_{\min}\}$. For the CIFAR-10 dataset we train a 100 layer DenseNet architecture as done in Huang et al. (2017), but we stop the training after 10 epochs instead of the original 300 used by the authors. The different configurations used for these experiments are summarized in Table 1. For instance, Figure 2 shows 1 realization of SGD training for DenseNet on CIFAR-100. In that figure we observe a rather stable trajectory, with not much oscillation. However, in Figure 8 we average over 2 realizations of SGD learning for DenseNet on CIFAR-10 and obtain highly oscillating trajectories. As expected, both trajectories follow a similar behavior as the $\alpha$-SMLC model.

In Figure 5(a) the same model as in Figure 3 is used but an additional regularization term is added to the loss function, that is $\lambda \|w\|_1$ where $w$ is a vector containing all the weights in the network. In Figure 5(b) a FCNN with a single hidden layer of size 100, and $tanh$ activations, is used and the dataset sizes are reduced according to 3. Other parameters are $\gamma_{\max} = 10^{-1}$, $\gamma_{\max} = 10^{-2}$, and the number of epochs is 10 000.

Table 3: Dataset Sizes for Figure 5(b)

| $p$ | Dataset Size |
|-----|--------------|
| 0.1 | 200 |
| 0.2 | 400 |
| 0.4 | 1000 |

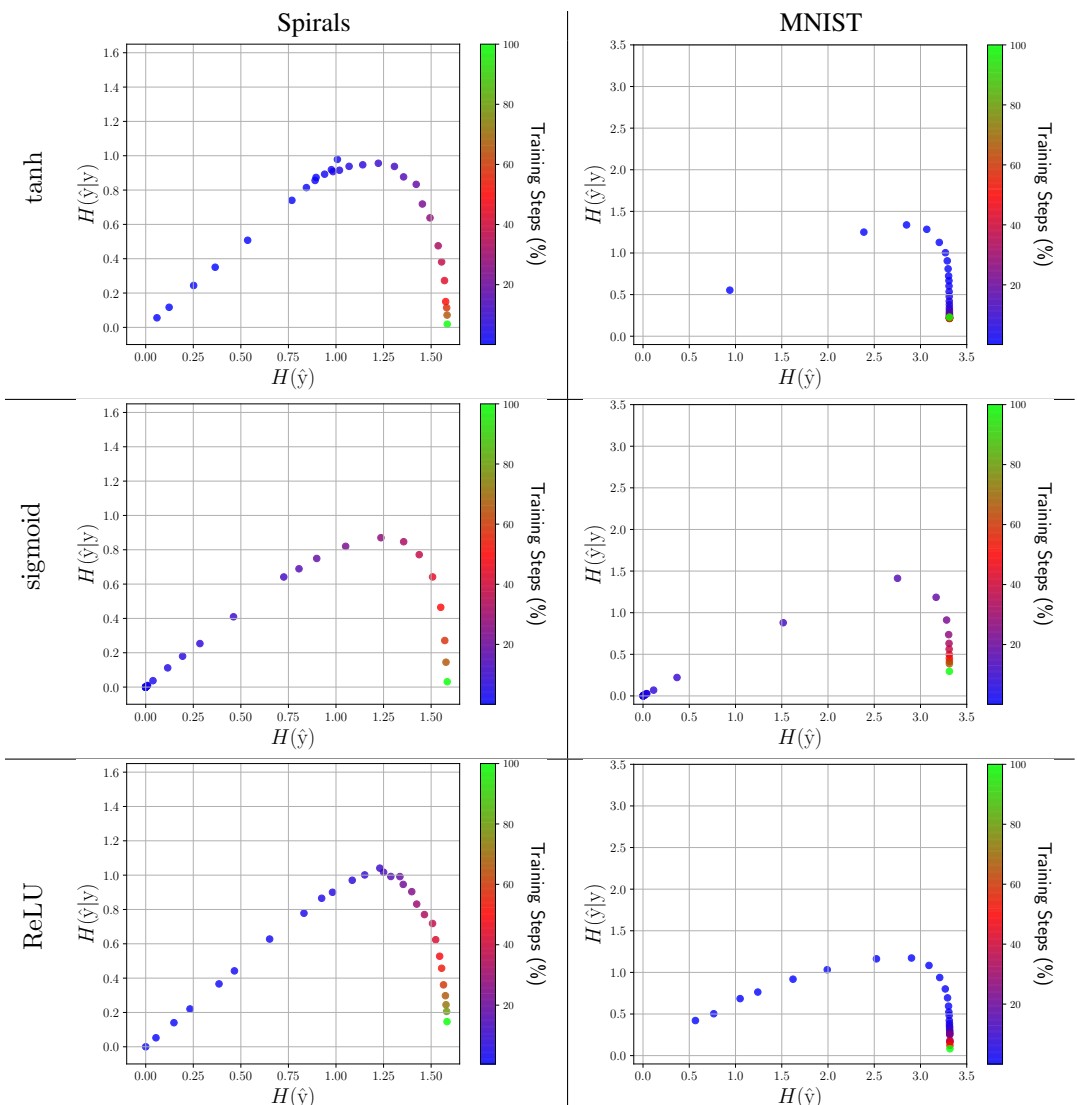

Figure 7: Information plane trajectory during the learning process (Anonymous, 2018).

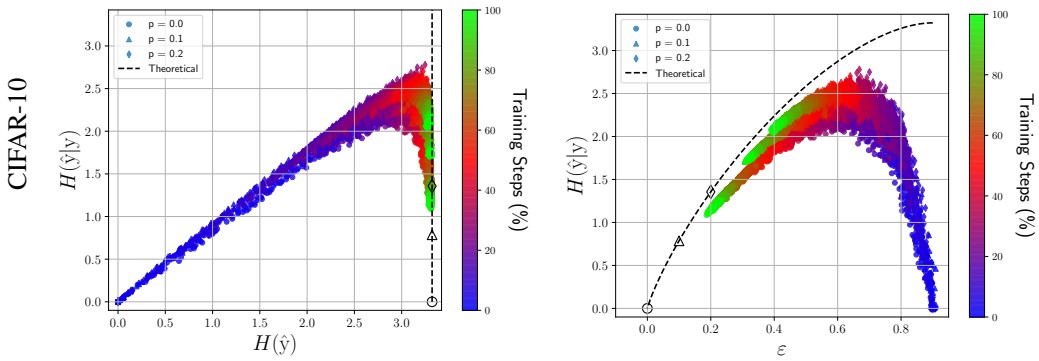

Figure 8: Experiments for DenseNet on CIFAR-10. This figures follow the same format as Figures 3 and 4.

