# OpenReview forum: "On the Trajectory of Stochastic Gradient Descent in the Information Plane"
_ICLR.cc/2019/Conference_

### Official Review · AnonReviewer2 · 2018-10-17
**Results of questionable value**

**Rating:** 2
**Confidence:** 4

**Review:**

The paper tries to describe SGD from the point of view of the distribution p(y',y) where y is (a possibly corrupted) true class-label and y' a model prediction. Assuming TV metric of probabilities, a trajectory is defined which fits to general learning behaviour of distributions.

The issue is that the paper abstracts the actual algorithm, model and data away and the only thing that remains are marginal distributions p(y) and conditional p(y'|y). At this point one can already argue that the result is either not describing real behavior, or is trivial. The proposed trajectory starts with a model that only predicts one-class (low entropy H(y') and high conditional entropy) and ends with the optimal model. the trajectory is linear in distribution space, therefore one obtains initially a stage where H(y') and H(y'|y) increase a lot followed by a stage where H(y'|y) decrease.

This is known to happen, because almost all models include a bias on the output, thus the easiest way to initially decrease the error is to obtain the correct marginal distribution by tuning the bias. Learning the actual class-label, depending on the observed image is much harder and thus takes longer. Therefore no matter what algorithm is used, one would expect this kind of trajectory with a model that has a bias.

It also means that the interesting part of an analysis only begins after the marginal distribution is learned sufficiently well. and here the experimental results deviate a lot from the theoretical prediction. while showing some parabola like shape, there are big differences in how the shapes are looking like.

I don't see how this paper is improving the state of the art, most of the theoretical contributions are well known or easy to derive. There is no actual connection to SGD left, therefore it is even hard to argue that the predicted shape will be observed, independent of dataset or model(one could think about a model which can not model a bias and the inputs are mean-free thus it is hard to learn the marginal distribution, which might change the trajectory)

 Therefore, I vote for a strong reject.

---

> ### Author Response · Authors · 2018-11-26
> **More discussions supporting the value and potential of the proposed experiments**
>
> We thank the reviewer for his/her comments. Valid concerns about the paper have been pointed out, which we aim to clarify in the latest version of the paper and the following comments:
>
>     1. “This is known to happen, because almost all models include a bias on the output.”
>
> Regarding the comment about the bias, first, it should be noted that learning the marginal first is not universal and depends on the learning strategy. For instance if only one class is learned at a time, then the marginal is learned along with the training. See the figures attached to the general comment above. Therefore the relation between bias and learning marginals is far from being trivial and heavily depends on the learning algorithm. This hints to our claim that understanding different learning trajectories in the information plane can be illuminating for analyzing the learning process.
>
>
>     2. “while showing some parabola like shape, there are big differences in how the shapes are looking like”
>
> Note that  $\alpha$-SMLC is a model used to explore how SGD is acting on probabilities during learning. Admitting its imperfection, similarities besides the parabolic shapes are notable. For example, the $\alpha$-SMLC predicts that the inflection point of  $H(\hat y | y)$ gets closer to the bound as $p$ increases, which is also observed in the experiments.
>
> Despite this, we share the reviewer’s concern and believe that elaborating sophisticated extensions of the $\alpha$-SMLC, that more closely resemble SGD, may be a fruitful research direction.
>
>     3. “There is no actual connection to SGD left”
>
> The connection to SGD arises from the fact that it can be modeled as a HMP. As far as the learning process can be modeled as a HMP, we expect the analysis to hold as well. However, SGD learning in ANNs is interesting as it ends up close to Fano’s curve (as predicted by the $\alpha$-SMLC) and as it is indicated in Appendix, it manages to learn the true labels from the noisy ones.
>
>
>     4. “one could think about a model which can not model a bias and the inputs are mean-free thus it is hard to learn the marginal distribution, which might change the trajectory”
>
> We are not sure what the reviewer means by this but we think this concern may be addressed in the first point (“generality”) of the general comment above. Note that a model that is not able to learn the marginal distribution violates assumption A3 from that comment.

---

### Official Review · AnonReviewer3 · 2018-11-07
**ICLR 2019 Conference Paper1465 AnonReviewer1**

**Rating:** 6
**Confidence:** 4

**Review:**

This paper study the trajectory of H(\hat{y}) versus H(\hat{y}|y) on the information plane for stochastic gradient descent methods for training neural networks. This paper was inspired by (Ziv and Tishby 17'), but instead of measuring the mutual information I(X;T) and I(Y:T), this paper proposed to measure H(\hat{y}) and H(\hat{y}|y), which are much easier to compute but carries similar meaning as I(Y;T) and I(X;T).

The interesting part of this paper appears in Section 4, where the author makes a connection between the SGD training process and \alpha-SMLC(strong Markov learning chain). SMLC is just simply linear combination of the initial distribution and the final stable distribution of the labels. The authors show that the trajectory of the real experiment is similar to that of SMLC.

Generally I think the paper is well-written and clearly present the ideas. Here are some pros and cons.

Pros 1: The trajectory presented in this paper is much more reliable than that in (Ziv and Tishby 17'), since measuring the entropy and conditional entropy of discrete random variables are much easier. Also it is easy for people to believe that the trajectory holds for various neural network structure and various activation functions.

Pros 2: The connection to SMLC is interesting and it may contain lot of insights.

Cons 1: One of my major concern is --- if you look at the trajectory of the experiment v.s. SMLC (Figure 3), they look similar at first glance. But if you look at it carefully, you will notice that the color of them are different! For SGD, the trajectory goes to the turning point very soon (usually no more than 10% of the training steps), whereas SMLC goes to the turning point much slower. How do the authors think about this phenomenon and what does this mean?

Cons 2: This paper is going to be more meaningful if the author can provide some discussions, especially about (1) what does the shape trajectory mean (2) what do the connection between the trajectory and Markov chain means (3) how can these connections be potentially useful to improve training algorithm? I understand that these questions may not be clearly answerable, but the authors should make this paper more inspiring such that other researchers can think deeper after reading this paper.

Cons 3: I suggest the authors using SGD instead of GD throughout the paper. Usually GD means true gradient descent, but the paper is talking about batched stochastic gradient descent. GD does not have Markovity.

Generally, I think the paper is on the borderline. I think the paper is acceptable if the author can provide more insights (against Cons 2).

---

> ### Author Response · Authors · 2018-11-26
> **More discussions + new experiment showing the potential of using the information plane**
>
> We would like to thank the reviewer for his/hers comments. We addressed the main concerns about the paper as follows:
>     1. “the trajectory of the experiment v.s. SMLC (Figure 3), they look similar at first glance. But if you look at it carefully, you will notice that the color of them are different!”
> The points are colored according to the % of training time, so it is dependent on the number of epochs considered. Although we conjecture that the trajectory of SGD should be the observed parabolic shape, the convergence speed can vary from model to model. Currently, $\alpha$-SMLC is parameterized linearly in $\alpha$ which is an arbitrary choice. With more general parametrization for $\alpha$, one can  get a different coloring. Therefore the colors do not play any significant role in the claim.
>
>     2. “(1) what does the shape trajectory mean (2) what do the connection between the trajectory and Markov chain means (3) how can these connections be potentially useful to improve training algorithm?”
> Please see the second point (“meaning”) from the general comment above.
>
>     3. “I suggest the authors using SGD instead of GD throughout the paper.”
> We have taken this comment into account and updated the paper accordingly.

---

### Official Review · AnonReviewer1 · 2018-11-07
**unclear motivation**

**Rating:** 4
**Confidence:** 3

**Review:**

In summary, this paper does the following:
- The initial problem is to analyze the trajectory of SGD in training ANNs in the space of  P of probability measures on Y \times Y. This problem is interesting, but difficult.
- the paper constructs a Markov chain that follows a shortest path in TV metric on P
(the \alpha SMLC)
- through experiments, the paper shows that the trajectories of SGD and \alpha-SMLC have  similar conditional entropy.

My issues with this paper are:
a/ The main result is a simulation. How general is this? Could it depend on the dataset? Could you provide some intuition or prove that for certain dataset, these two trajectories are the same (or very close)?
b/ Meaning of this trajectory. This is not the trajectory in P, it is the trajectory of the entropies. In general, is there an intuitive explanation on why these trajectories are similar? And what does it mean -- for example, what would be a possible implication for training SGD? Could it be that all learning methods will have this characteristic parabolic trajectory for entropies?
c/ The theoretical contribution is minor: both the techniques and results quoted are known.

Overall, I think the paper lacks a take-away. It is an interesting observation that the trajectory of \alpha-SMLC  is similar to that of SGD in these plots, but the authors have not made a sufficient effort to interpret this.

---

> ### Author Response · Authors · 2018-11-26
> **New experiment showing that the information plane carries useful information about the training process that is hidden in train/test error.**
>
> We would like to thank the reviewer for his/hers comments, which lead us to improve our paper. Since these comments are shared with other reviewers we have posted the in a general comment above. Here is a summary of the main concerns of the reviewer:
>     1. “How general is this?”
>
> Please see the first point (“generality”) from the general comment above.
>
>     2. “Meaning of this trajectory.”
>
> Please see the second point (“meaning”) from the general comment above.
>
>     3. “I think the paper lacks a take-away.”
>
> Please see the second point (“meaning”) from the general comment above.

---

### Author Response · Authors · 2018-11-26
**General Comments for all Reviewers**

We thank all reviewers for their comments. Here we address some concerns about the paper that are common among the reviews:

1. (Generality) How General is this trajectory?

An important motivation of this work is to explore the generality of the observed trajectories and its interpretation.

We have tried to show that as far as some assumptions are in place, some features of the trajectory are to be expected in general:

A1) The labels Y are uniformly distributed.
A2) The parameter updates during learning are done using a function of the previous parameters and an independent random variable representing the training batch, i.e.,   $\theta_{n+1} = f(\theta_{n}, U)$. This assumption holds for SGD training of ANNs.
A3) We assume perfect learning, that is $g(\theta_n, .)$ converges to $c(.)$.

With these assumptions, the trajectory of learning in the information plane is independent from the architecture of the classifier (which may not even be an ANN).

First, since the output labels are uniformly distributed, the entropy $H(\hat y)$ tends to increase to its maximum during successful training which is motivated by Proposition 3.

Understanding learning as maximization of mutual information, the shape of $H(\hat y | y)$ is also expected to consist of one or more bumps with local maxima in the middle. The number of bumps depends on the trajectory and can be different if the learning strategy is different. For instance, if the learning algorithm is devised to learn one label at a time, the trajectory is quite different as it is shown in the Figure (here: https://ibb.co/f9H4TzP ) for 3 and 10 classes.

The parabola shape of the conditional entropy, however, is conjectured to be due to SGD changing the probability of labels to the ground truth labels by moving on the shortest path on the space of joint probability measures. This claim requires more theoretical and experimental investigation and it is relegated to future works.

2. (Meaning) What is the meaning of these trajectories?

We propose that this trajectory in the information plane carries many useful information about the training process and can be used effectively to observe the state of training beyond mere measurement of training/test error. To back up this claim we included a new experiment in the paper in which we use the trajectories to spot underfitting and overfitting.

A common practical issue in training learning algorithms is to find roots of the low accuracy and see if the obtained accuracy is the best we can do. This can be done in a straightforward way using the information plane.

We show that, regardless of the error, an underfitted classifier lies far from Fano’s bound which implies that the accuracy can be still improved. On the other hand, an overfitted classifier lies very close to the Fano bound and moves on this curve. From this experiment, we conclude that taking a look at the trajectory of these quantities in other scenarios (e.g. adversarial training) may reveal characteristic behaviors thus providing new insights.

---

### Meta-Review · Area_Chair1 · 2018-12-18
**New approach to monitor learning, more work needed to clarify meaning and potential use**

**Confidence:** 4
**Recommendation:** Reject

**Metareview:**

The paper proposes a quantity to monitor learning on an information plane which is related to the information curves considered in the bottleneck analysis but is more reliable and easier to compute.

The main concern with the paper is the lack of interpretation and elaboration of potential uses. A concern is raised that the proposed method abstracts away way too much detail, so that the shapes of the curves are to be expected and contain little useful information (see AnonReviewer2 comments). The authors agree to some of the main issues, as they pointed out in the discussion, although they maintain that the method could still contain useful information.

The reviewers are not very convinced by this paper, with ratings either marginally above the acceptance threshold, marginally below the acceptance threshold, or strong reject.